# Blood Circulating CD133+ Extracellular Vesicles Predict Clinical Outcomes in Patients with Metastatic Colorectal Cancer

**DOI:** 10.3390/cancers14051357

**Published:** 2022-03-07

**Authors:** Davide Brocco, Pasquale Simeone, Davide Buca, Pietro Di Marino, Michele De Tursi, Antonino Grassadonia, Laura De Lellis, Maria Teresa Martino, Serena Veschi, Manuela Iezzi, Simone De Fabritiis, Marco Marchisio, Sebastiano Miscia, Alessandro Cama, Paola Lanuti, Nicola Tinari

**Affiliations:** 1Department of Pharmacy, University “G. D’Annunzio” Chieti-Pescara, 66100 Chieti, Italy; laura.delellis@unich.it (L.D.L.); serena.veschi@unich.it (S.V.); cama@unich.it (A.C.); 2Department of Medicine and Aging Sciences, University “G. D’Annunzio” Chieti-Pescara, 66100 Chieti, Italy; pasquale.simeone@unich.it (P.S.); bucadavide@gmail.com (D.B.); simone.defabritiis@unich.it (S.D.F.); marco.marchisio@unich.it (M.M.); sebastiano.miscia@unich.it (S.M.); 3Center for Advanced Studies and Technology (C.A.S.T.), University “G. D’Annunzio” Chieti-Pescara, 66100 Chieti, Italy; m.iezzi@unich.it; 4Clinical Oncology Unit, S. S. Annunziata Hospital, 66100 Chieti, Italy; pietro.dimarino@unich.it (P.D.M.); mtmartino@virgilio.it (M.T.M.); 5Department of Innovative Technologies in Medicine and Dentistry, University “G. D’Annunzio” Chieti-Pescara, 66100 Chieti, Italy; detursi@unich.it (M.D.T.); grassa@unich.it (A.G.); 6Department of Neurosciences, Imaging and Clinical Sciences, University “G. D’Annunzio” Chieti-Pescara, 66100 Chieti, Italy; 7Department of Medical, Oral & Biotechnological Sciences, University “G. D’Annunzio” Chieti-Pescara, 66100 Chieti, Italy; ntinari@unich.it

**Keywords:** extracellular vesicles, circulating biomarkers, colorectal cancer

## Abstract

**Simple Summary:**

In this study, we explored the prognostic and predictive value of blood circulating EVs expressing selected surface proteins in patients with metastatic colorectal cancer (mCRC). A recently patented flow cytometry protocol was used for the identification and subtyping of blood circulating EVs in a cohort of patients with stage IV colorectal cancer (*n* = 54) and in a cohort of healthy controls (*n* = 48). We observed an increased blood concentration of tumor-induced blood circulating EVs in the mCRC cohort as compared to healthy controls. Additionally, we show an intriguing link between circulating CD133+ EVs and poor clinical outcomes in patients with mCRC. This study provides novel insights about the potential impact of EVs as a relevant source of candidate biomarkers in mCRC.

**Abstract:**

Colorectal cancer (CRC) is one of the most incident and lethal malignancies worldwide. Recent treatment advances prolonged survival in patients with metastatic colorectal cancer (mCRC). However, there are still few biomarkers to guide clinical management and treatment selection in mCRC. In this study, we applied an optimized flow cytometry protocol for EV identification, enumeration, and subtyping in blood samples of 54 patients with mCRC and 48 age and sex-matched healthy controls (HCs). The overall survival (OS) and overall response rate (ORR) were evaluated in mCRC patients enrolled and treated with a first line fluoropyrimidine-based regimen. Our findings show that patients with mCRC presented considerably higher blood concentrations of total EVs, as well as CD133+ and EPCAM+ EVs compared to HCs. Overall survival analysis revealed that increased blood concentrations of total EVs and CD133+ EVs before treatment were significantly associated with shorter OS in mCRC patients (*p* = 0.001; and *p* = 0.0001, respectively). In addition, we observed a correlation between high blood levels of CD133+ EVs at baseline and reduced ORR to first-line systemic therapy (*p* = 0.045). These findings may open exciting perspectives into the application of novel blood-based EV biomarkers for improved risk stratification and optimized treatment strategies in mCRC.

## 1. Introduction

Colorectal cancer (CRC) is one of the leading causes of cancer-related morbidity and mortality in Western countries [1]. Almost 20% of patients with CRC are metastatic at diagnosis and approximately 25% of those treated for localized disease will later develop metastases [2]. Treatment advances in metastatic colorectal cancer (mCRC) led to a significant increase in patient overall survival (OS) [3]. More effective systemic therapies have been introduced, including chemotherapy, targeted therapy, immunotherapy, and combinations of these treatment strategies [4]. Moreover, prognostic and predictive biomarkers are crucial to identify high-risk patients, optimize clinical management and personalize therapeutic approaches in mCRC. Currently, treatments in mCRC are tailored according to tumor molecular profiling (e.g., K-RAS/N-RAS/B-RAF and MMR status) [5]. However, tissue-based biomarkers may not represent the overall tumoral heterogeneity and are obtained by invasive as well as time-consuming procedures. Conversely, liquid biopsy is emerging as a rapid and easily accessible alternative or complementary procedure to tissue biopsy [6]. Specifically, this approach relies on the detection of tumor-associated components, including circulating tumor cells, circulating tumor DNA and RNA, extracellular vesicles (EVs) as well as tumor-educated cells in biological fluids of cancer patients [6]. In this light, liquid biopsy paves the way for improved personalization of CRC management [7].

Extracellular vesicles (EVs) are lipid membrane-enclosed, nanosized particles secreted by cells to regulate intercellular communication. EVs are involved in most steps of cancer progression [8]. Accumulating evidence show an EV key role in promoting tumor growth, formation of premetastatic niche, neoangiogenesis, disruption of the blood–brain barrier, drug resistance, and immune cancer evasion [9]. A unique biomolecular cargo characterizes tumor-associated EVs, which steadily circulate in almost all kinds of body fluids [10,11]. Therefore, circulating EVs are considered valuable carriers of tumor information and promising biomarkers for cancer diagnosis, prognostication, and surveillance [12,13,14]. In this regard, the analysis of blood circulating EV cargo revealed cancer-specific molecules including proteins, microRNAs, long noncoding RNAs, and circular RNAs that may be employed for CRC early detection and risk stratification [15]. However, little is known regarding the prognostic and predictive role of circulating EV subpopulations characterized by EV surface protein expression in patients with CRC. EV surface proteins regulate EV targeting, protect EVs from blood clearance and control EV capture by recipient cells [16]. Therefore, the analysis of EV phenotypes may provide key information regarding the origin and the donor cell types, thus suggesting fate and functions of blood circulating EVs [17]. Nevertheless, wide protein heterogeneity of blood circulating EV surfaceome and over-representation of certain EV subpopulations in blood samples could limit the identification of rare and specific EV subtypes by applying bulk techniques for protein identification such as Western blotting and enzyme-linked immunosorbent assay (ELISA) [18]. Thus, highly sensitive characterization of blood-derived EV subtypes that requires small sample volume and minimal pre-analytical sample processing is needed for accelerating clinical translation of EV research findings [19]. In this respect, recent reports highlighted the clinical potential of rapid as well as sensitive phenotypical profiling of blood circulating EVs in patients with CRC [20,21].

Here, we employed a recently patented polychromatic flow cytometry (PFC) method to characterize intact blood circulating EVs expressing putative cancer markers in patients with advanced CRC. Circulating tumor-associated EVs were characterized according to CD133 and EPCAM expression. Surface EV markers were selected in line with previous preclinical and clinical CRC studies, which analyzed CD133 and EPCAM expression for EV characterization [22,23,24,25,26,27]. We compared EV concentrations between CRC patients and healthy controls and correlated EV blood levels with patient clinical-pathological characteristics. Moreover, we investigated the prognostic and predictive value of total as well as CD133+/ and EPCAM+ circulating EVs in treatment-naïve patients with metastatic CRC.

## 2. Materials and Methods

### 2.1. Patients

This prospective observational study enrolled adult patient candidates for systemic treatment, with a histologically or cytologically confirmed diagnosis of stage IV colorectal cancer. Patients were recruited from the Clinical Oncology Unit of the SS Annunziata Hospital in Chieti (Italy) from January 2017 to August 2021. Baseline demographic and clinical characteristics of all enrolled mCRC patients (*n* = 54) were summarized in Appendix A. A cohort of age and sex-matched healthy controls was also included in the study (*n* = 48).

All procedures involving human participants were carried out in accordance with the ethical standards of the 1964 Helsinki declaration and its later amendments or with comparable ethical standard. This study was approved by the local ethics committee on 25 February 2016. All patients gave a written informed consent. 

### 2.2. Blood Collection 

A baseline peripheral blood sample was collected at the time of enrollment and an additional blood sample was drawn at 12 (+/−6) weeks after cycle 1 of therapy. Peripheral blood was collected (21 G needles) in two sodium citrate tubes (Becton Dickinson Biosciences-BD, San Jose, CA, USA). Peripheral blood samples were processed within 4 h from venipuncture. The first harvested tube was discarded to minimize the EV release induced by vascular damage effects.

### 2.3. Flow Cytometry Detection of Extracellular Vesicles

Lipophilic cationic dye (LCD) and reagents reported in Appendix A were mixed in a PBS solution and then 5 μL of whole blood were added. Each reagent stock solution was centrifuged before its use (21,000× *g*, 10 min) to exclude immune complex formation and the unspecific background caused to the antibody aggregation. After 45 min of staining at room temperature in the dark, the sample was appropriately diluted with PBS 1× and 1 × 10^6^ events/sample were acquired by flow cytometry. Extracellular vesicle concentrations were obtained by volumetric count (FACSVerse, BD Biosciences, San Jose, CA, USA). The trigger threshold was set up on the appropriate fluorescent channel (APC), as previously described [28,29]. The signal pulse height (H) was used for forward scatter (FSC), side scatter (SSC), and any fluorescent signals. EV scatter gates were defined and verified overtime by using both the Megamix-Plus beads (Byocitex, Marseille, France) and the Rosetta Calibration System (Exometry, Amsterdam, The Netherlands), as previously reported [28,29]. The gating strategy and the evaluation of non-specific fluorescence was determined by fluorescence minus one (FMO) controls combined with the appropriate isotype control [30]. Reagent-only, buffer-only, and 1% Triton X-100 controls were also acquired to correctly discriminate the EV population from contamination and debris. Compensation was assessed using CompBeads (BD) and single-stained fluorescent samples. Data were analyzed using FACSuite v 1.0.6.5230 (BD) and FlowJo v 10 (BD Biosciences). 

### 2.4. Flow Cytometry Subtyping of Extracellular Vesicles 

The whole population of active and intact EVs was identified as LCD+/phalloidin− events, falling in the scatter area containing events with physical parameters lower than platelets (Figure 1A(a)). The size and morphological characteristics of EVs detected by this LCD-based protocol were detailed in previous reports [28,29]. Marchisio et al. reported that more than 90% of blood circulating LCD+/phalloidin-particles presented a diameter larger than 160 nm. Total extracellular vesicles (LCD+/phalloidin− events) were analyzed on a CD45-H/CD133-H dotplot (Figure 1A(b)) and CD45− events were gated (Figure 1A(c)). The CD45− population was analyzed on a CD326-H/CD133-H dotplot and CD133+ (CD45−/CD133+ events) and EPCAM+ (CD45−/EPCAM+ events) EVs were identified (Figure 1A(d,e)). 

### 2.5. Statistical Analysis 

Statistical analysis was performed using SPSS v25.0 (IBM SPSS, Chicago, IL, USA) and Medcalc v14.8.1 (MedCalc Software, Ostend, Belgium). No assumption of normality of the data was formulated; therefore, non-parametric tests were employed for comparisons. Continuous variables were compared using the Mann–Whitney *U* test. Correlations between EV concentrations and clinical-pathological variables were calculated by Spearman’s rank correlation coefficients. Univariate and multivariate Cox proportional hazards models were applied to calculate the hazard ratio (HR) together with 95% confidence intervals (CIs). EV concentrations were analyzed as continuous variables to maximize the statistical power of the univariate and multivariate analysis [31]. Log-transformation was applied to the EV concentration to reduce skewness in its distribution. EV concentrations were dichotomized according to survival outcome utilizing the Charité Cutoff Finder functions [32]. Median overall survival (mOS) was computed using the Kaplan–Meier (KM) curve estimator. Differences in mOS were evaluated using the log-rank test. The overall response rate (ORR), defined as the percentage of patients who achieved complete response (CR) and partial response (PR), was analyzed and used to identify the responder and non-responder cohorts. ORRs were compared using Fisher’s exact test. The ROC curve of response (CR + PR) vs. non-response (SD + PD) was calculated to assess the predictive ability of pre-treatment EV concentrations. The relative EV count change was calculated as % change ({[log2EV concentration week 12/log2EV concentration week 0] − 1} ∗ 100) and divided into two groups: decreasing (≥25% decrease) and stable or increasing (<25% decrease to <25% increase or ≥25% increase) EV concentration. The SPSS biased-corrected and accelerated bootstrap method with 1000 bootstrap samples and a 95% confidence interval was applied for internal validation. A *p*-value of <0.05 was considered statistically significant.

## 3. Results

### 3.1. Patients with Metastatic CRC Present Higher Levels of Total and CD133+ Blood Circulating EVs as Compared to Healthy Controls

Blood concentrations of total, CD133+ and EPCAM+ EVs were determined in a cohort of patients with colorectal cancer (*n* = 54) and in a group of age and sex-matched HCs (*n* = 48). Median blood concentrations at baseline of total, CD133+, as well as EPCAM+ EVs were compared between patients and HCs, as reported in Table 1 and Figure 1B. Blood concentrations of both total and the analyzed EV subtypes were significantly higher in mCRC patients as compared to HCs (median total EVs/μL (95% CI) = 5264.0 (4123.0–6314.0) vs. 2548.0 (2100.7–3051.4), respectively; median CD133+ EVs/μL (95% CI) = 52.6 (32.4–96.1) vs. 18.4 (11.6–32.2), respectively; median EPCAM+ EVs/μL (95% CI) = 50.9 (38.6–67.2) vs. 27.0 (19.3–42.0), respectively (Table 1)). 

Correlations between blood circulating EVs and clinical-pathological factors including age, sex, Eastern Cooperative Oncology Group (ECOG) performance status (PS), primary tumor location, tumor grading, K-RAS mutation status, liver metastasis, lung metastasis, and number of metastatic sites were explored in the overall patient cohort (Appendix A). However, blood concentrations of total and the analyzed EV subtypes did not correlate with any of the clinical-pathological factors included in the analysis.

In the mCRC cohort, blood EV concentrations were compared between pre-treatment and post-treatment samples. Pre-treatment samples included those collected at baseline in treatment-naïve patients (*n* = 36). Post-treatment EV concentrations referred to those analyzed in samples collected after or during the first line treatment (*n* = 41). Blood levels of total EV and CD133+ EV concentrations tended to decrease during therapy, although no significant difference in total and the analyzed EV subtype concentrations was observed between the pre- and post-treatment conditions (Appendix A). 

### 3.2. Total and CD133+ EVs Concentrations Are Associated with Overall Survival in Treatment Naïve Patients

We examined whether EV levels at baseline were correlated with survival in a cohort of treatment naïve patients. Patients enrolled before a first-line treatment and treated with a fluoropyrimidine-based chemotherapy regimen were included in the survival analysis (*n* = 33). Median follow-up time was 11.0 (95% CI 8.0–19.0) months and 24 (72.7%) patients were alive at the time of the analysis. The overall one-year OS was 81%.

We performed univariate and multivariate Cox proportional hazards regression analysis to evaluate the association between pre-treatment EV concentration and survival. Univariate Cox proportional hazards regression analysis was used to assess different variables including ECOG PS, age, number of metastatic sites, tumor grading, and primary tumor site as potential risk factors affecting OS. Those factors resulting significantly correlated with OS (*p* < 0.05) in the univariate analysis were included in the multivariable Cox regression model.

Univariate analysis showed that higher total EV concentrations were associated with a significant increase in the risk of death (HR (95% CI) = 1.77 (1.24–2.54)) (*p* = 0.002) (Table 2). Similarly, higher blood levels of CD133+ EVs was correlated with a worse survival (HR (95% CI) = 1.72 (1.24–2.39)) (*p* = 0.001) (Table 3). Conversely, the risk of death did not significantly differ according to the EPCAM+ EV concentration (Table 3). Multivariate analysis indicated that higher total EVs and CD133+ EV concentrations were independently associated with a lower survival probability (Table 3). Univariate and multivariate analyses were confirmed via bootstrap validation. 

The relationship between OS and total EV and CD133+ EV blood concentrations was also evaluated using Kaplan–Meier curves (Figure 2). We dichotomized the EV levels for survival analysis using cut-off values calculated with the Charité Cutoff Finder tool (Total EVs cut-off = 7007 EVs/μL; CD133+ EVs cut-off = 163.6 EVs/μL), as reported in Methods (Figure 2A(b),B(b)). Of note, patients presenting higher total EV concentration (>7007 EVs/μL) were characterized by a considerably unfavorable outcome with a mOS of 9 months (95% CI 6.1–11.9) as compared to a mOS not reached at the time of analysis cut-off in the group of patients with a lower EV concentration (*p* = 0.001) (Figure 2A(a)). Similarly, mOS was significantly shorter in the group with higher CD133+ EV concentrations (>163.6 EVs/μL) than in patients with lower blood levels of CD133 expressing EVs (*p* = 0.0001) (Figure 2B(a)). We also evaluated the correlation between OS and CD133+ EV concentration according to K-RAS mutation status (Appendix A). In this regard, we observed that higher CD133+ EV concentration was associated with worse mOS in both the K-RAS mutated and K-RAS wild-type cohorts of patients (Appendix A). 

### 3.3. Blood Circulating CD133+ EVs Are Associated with Overall Response Rate

We then explored whether blood circulating EV concentrations correlated with treatment response in the same group of patients included in the survival analysis. An objective response was evaluable in 30 out of 33 patients. The overall response rate in the whole cohort (*n* = 30) was 46.7%. Median PFS was 12.0 months (95% CI 7.5–16.5).

The blood concentration of EVs at baseline was compared between responders and non-responders (Figure 3; Appendix A). Of note, responders presented a significantly lower median pre-treatment concentration of CD133+ EVs (*p* = 0.03) (Figure 3A). Conversely, no significant differences in total EV and EPCAM+ EV concentrations were reported between responders and non-responders (Appendix A). As shown in Figure 3C, the receiving operator characteristic (ROC) curve analysis confirmed a correlation between treatment response and CD133+ blood circulating EVs (AUC = 0.728 [CI 95% 0.542–0.913]; *p* = 0.03). We then compared the overall response rate (ORR) between the group with higher CD133+ EV concentrations (>163.6 EVs/μL) and the group with lower CD133+ EV concentrations (≤163.6 EVs/μL). The overall response rate was lower in patients with high as compared to those with low CD133+ EVs (20.0 vs. 60.0%, respectively; *p* = 0.045) (Figure 3B). Additionally, we analyzed differences in progression-free survival (PFS) between the two groups of patients with different blood levels of CD133 + EVs. We observed that PFS in the high CD133+ EV group was significantly shorter as compared to the low CD133+ EV group (*p* = 0.0003) (Figure 3D).

In addition, the association between treatment response and variations of EV concentrations during therapy was examined. On-treatment EV concentration was available for 18 of 30 patients. In this regard, responders and non-responders were equally distributed between the two groups of stable or increasing and decreasing total EVs, as well as CD133+ and EPCAM+ EV concentrations (Appendix A).

## 4. Discussion

CRC is one of the leading causes of cancer-related death worldwide. Despite recent advances in CRC diagnosis, management and treatment, there are still few CRC-related biomarkers used in clinical practice. In this regard, the application of blood-based biomarkers is of great interest because they are minimally invasive and may provide dynamic information about the overall cancer phenotypical and functional state [6]. The analysis of blood circulating EVs represents an emerging form of liquid biopsy in patients with cancer. Cancer-associated EVs may reach the bloodstream, becoming valuable circulating carriers of tumor-induced molecules. Thus, blood-derived EVs represent attractive circulating biomarkers to profile the CRC dynamics and improve CRC early diagnosis, prognostication, as well as prediction of treatment response [7]. In the present study, we used a novel optimized protocol using polychromatic flow cytometry (PFC) to characterize blood circulating EVs expressing putative CRC markers and explore their prognostic as well as predictive value in patients with stage IV colorectal cancer.

We report herein a considerable increase in blood circulating extracellular vesicle concentration in patients with mCRC as compared to healthy controls (HCs). This observation is in line with previous studies that reported an association between high levels of blood circulating extracellular vesicles and cancer, including colon cancer [23,25,33,34,35]. The expansion of the EV blood compartment can be related both with increased EV release by cancer cells and with enhanced tumor-induced EV secretion in non-malignant cells [10]. However, the understanding of tumor-related biological information carried by the overall circulating EV burden largely depends on the characterization of EV phenotype and cargo [12]. Accordingly, in this study, we aimed to characterize the expression of putative tumor markers including CD133 and EPCAM in blood circulating EVs of patients with metastatic CRC. Of note, we observed that patients presented considerably higher levels of CD133+ and EPCAM+ EVs as compared to HCs. CD133 and EPCAM are transmembrane proteins that are typically overexpressed in colon cancer [36]. Previous in vitro studies showed that cancer cells can release CD133− and EPCAM-enriched EVs [22,37]. Tumor-associated EVs extensively circulate in biological fluids of cancer patients, and it is conceivable that high levels of blood-derived CD133+ and EPCAM+ EVs in patients with mCRC might be related with an increased secretion of these EV subtypes by cancer cells. Nevertheless, there is evidence that colorectal expression of CD133 and EPCAM can be induced by both CRC- and non-CRC-related inflammation [38,39]. In addition, Bobinger et al. reported that enhanced levels of CD133+ particles can be detected in biological fluids of patients affected by inflammatory diseases [40]. Thus, it should be considered that tumor-related inflammation and increased cell turnover could favor cancer-induced secretion of CD133+ and EPCAM+ EVs by non-malignant cells. All these factors may affect blood levels of circulating CD133+ and EPACM+ EVs in patients with CRC.

In this study, survival analysis revealed that high blood levels of total EVs were significantly correlated with worse survival in a cohort of treatment naïve patients with advanced CRC. In this regard, it has been reported that high levels of blood circulating extracellular vesicles correlate with low overall survival in patients with cancer [41,42]. Higher concentrations of circulating EVs were considered to be associated with an increased release of cancer-derived EVs, supporting several pro-tumor functions including cancer progression, metastasis, treatment resistance, and immune escape [33,43]. Thus, increased levels of EVs may reveal the presence of tumors with more aggressive biology, which could negatively affect patient survival. Nonetheless, the prognostic value of total circulating EVs can be influenced by the amount of EVs derived from non-tumor cells. In this respect, previous studies showed a correlation between increased blood circulating platelet or endothelial-derived EVs and reduced survival in patients with advanced cancer [13,44,45]. In this regard, it is crucial to understand the specific contribution of single EV subpopulations to the biological and clinical significance of the overall blood EV compartment in patients with cancer. Thus, it is necessary to characterize blood circulating EV phenotype. In this report, survival analysis showed that circulating EVs characterized by CD133 expression presented an independent prognostic value in patients with advanced CRC. Conversely, the blood concentration of EPCAM+ EVs was not associated with OS. Specifically, we observed that higher levels of circulating CD133+ EVs were correlated with shorter OS. We previously reported a similar correlation between blood concentration of CD133+ EV subpopulations and survival in a cohort of advanced non-small cell lung cancer [23]. Furthermore, tumor expression of CD133 was previously shown to be associated with recurrence, metastasis, and poor survival in CRC [46]. In addition, in vitro studies indicated that CD133-enriched EVs directly promoted tumor growth, metastasis, cancer cell proliferation, and motility in various cancer types, including CRC [22,47]. It should be noted that CD133 is recognized as an effective colon cancer stem cell (CSC) marker [48]. CSC-derived EVs were shown to have a role in inducing stem-like properties in differentiated cancer cells, transferring chemotherapy resistance and promoting metastasis as well as angiogenesis [49]. Thus, enhanced secretion of tumor-associated EVs can be related with tumor progression, suggesting a potential explanation concerning the negative prognostic value of circulating CD133+ EVs that we described in patients with advanced CRC [50]. In our report, blood concentration of CD45− EPCAM+ EVs did not significantly correlate with OS. EPCAM and CD45 have been largely employed to enrich circulating tumor cells (CTCs), which showed prognostic potential in CRC [51]. In this regard, Nanou et al. explored the prognostic value of CTC and tumor-derived extracellular vesicles (TdEV) in a large cohort of patients with metastatic CRC [27]. Specifically, TdEV were identified as blood circulating EpCAM+, cytokeratins (CK)+, 4,6-diamidino-2-phenylindole (DAPI)−, CD45− microparticles. In this study, high levels of TdEVs and CTCs were both significantly associated with worse overall survival. These findings suggest that further phenotypical characterization of EPCAM+ circulating EVs may be helpful for improved detection of tumor associated EVs to employ as prognostic tool in CRC.

We observed that blood-derived CD133+ EVs were significantly correlated with response to systemic therapy. Specifically, pre-treatment higher concentrations of blood circulating CD133+ EVs were associated with reduced ORR and shorter PFS in patients with advanced CRC treated with a fluoropyrimidine-based chemotherapy regimen. In this regard, several lines of evidence reported a correlation between drug resistance and the tumor expression of CD133 [52,53]. In detail, CD133 mediated the upregulation of PI3K and FLIP3 protein pathways in cancer cells, preventing apoptosis as well as autophagy and thus promoting chemoresistance [54,55]. Moreover, results from two in vitro studies showed that CD133-enriched EVs released by colon cancer cells induced drug resistance by activating signaling pathways in recipient cells [22,26]. Intriguingly, Kang M. et al. reported that colon cancer cells harboring wild-type KRAS develop resistance to anti-EGFR agents after treatment with cancer cell-derived CD133+ microvesicles [22]. It should be noted that a group of patients included in our analysis were treated with anti-EGFR drugs in combination with chemotherapy. Thus, preclinical studies support an active role of CD133+ EVs in promoting CRC cell chemoresistance and this would be in line with our findings. In summary, we observed an intriguing correlation between increased blood concentrations of CD133+ EVs and treatment failure in mCRC, although larger clinical studies and further preclinical evidence will be needed to validate these results.

## 5. Conclusions

In this study, PFC-based immunophenotypical characterization of circulating EVs indicated that blood levels of total, CD133+ and EPCAM+ EVs are higher in patients with advanced CRC as compared to healthy controls, indicating that this increase reflects phenotypic changes induced by the tumor. Additionally, this study provides evidence that increased blood concentration of CD133+ EV is associated with poor survival and reduced response to systemic therapy in advanced CRC. This intriguing observation points at potential mechanisms modulating response to therapy, which will need to be further investigated.

In conclusion, we provide novel insights into the prognostic and predictive potential of circulating tumor-induced EVs which may foster the use of liquid biopsy for improved personalized medicine in stage IV CRC.

## Figures and Tables

**Figure 1 cancers-14-01357-f001:**
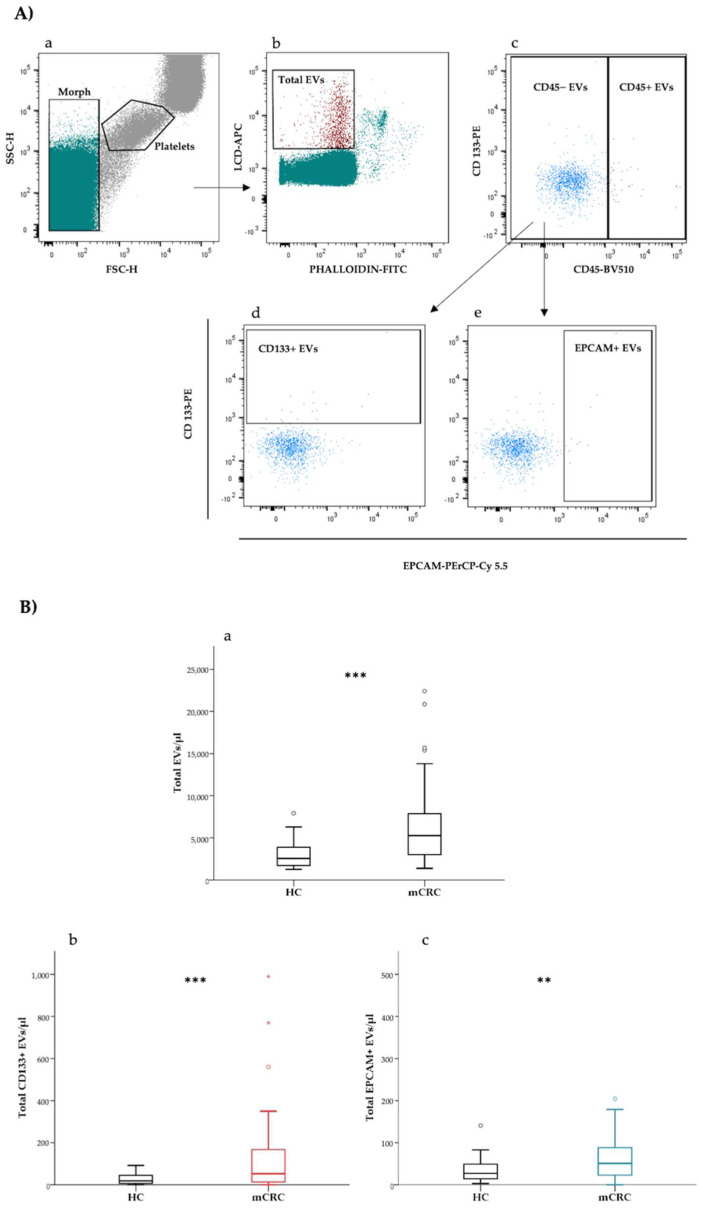
(**A**) Gating strategy for the identification and subtyping of extracellular vesicles (EVs) in peripheral blood samples. All events were represented on a forward scatter-H/side scatter-H dot-plot and a “platelet-free area” (Morph) region was defined by using platelets as a reference population: (**a**) the “platelet-free area”(Morph) was shown on a phalloidin-H/lipophilic cationic dye (LCD)-H dot-plot and total EVs were identified as LCD-positive/phalloidin negative events. (**b**) Total EVs were analyzed on a CD133-H/CD45-H dot-plot and CD45− and CD45+ events were gated (**c**). The CD45− population was plotted on a CD133-H/EPCAM-H dot-plot and CD133+ EVs (CD45−/CD133+ events) (**d**) and EPCAM+ EVs (CD45−/EPCAM+ events) (**e**) were identified. (**B**) Box plots showing differences in total EV (**a**), CD133+ EV (**b**) and EPCAM+ EV (**c**) concentration between patients with mCRC (*n* = 54) and healthy controls (*n* = 48). **, *p* < 0.01; ***, *p* < 0.001. Extreme values were not shown.

**Figure 2 cancers-14-01357-f002:**
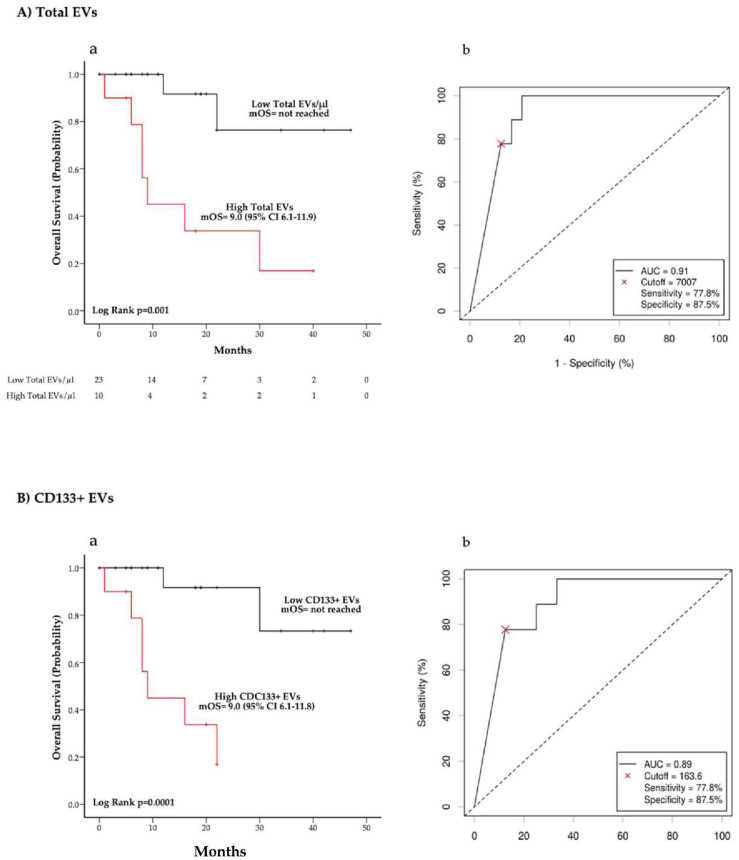
Kaplan–Meier (KM) curves showing the relationship between overall survival and blood concentration of total EVs and CD133+ EVs are illustrated, respectively, in panel (**A**(**a**)) and (**B**(**a**)). ROC curves for identification of optimal cut-off points are shown in panel (**A**(**b**)) and panel (**B**(**b**)).

**Figure 3 cancers-14-01357-f003:**
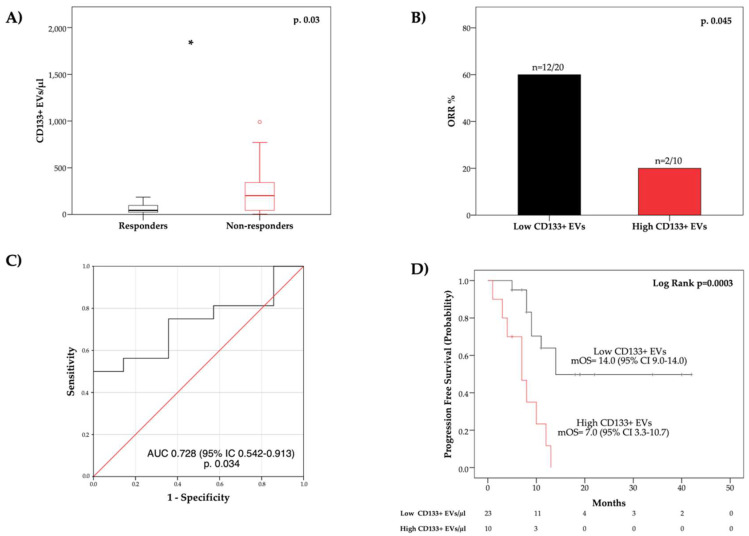
Relationship between treatment response and blood circulating CD133+ EV concentration at baseline. (**A**) Box plot diagram showing difference in blood concentration of CD133+ EVs between responders and non-responders are represented. (**B**) Histograms illustrate overall response rate in patients with high and low CD133+ EVs; *, *p* < 0.05. (**C**) Receiver operating characteristic (ROC) curve showing the effect of CD133+ EVs in predicting treatment response are shown. (**D**) Kaplan–Meier curves comparing PFS between two groups of patients with different blood concentration of CD133+ EVs are represented.

**Table 1 cancers-14-01357-t001:** Comparison of total and subtype EV concentrations between mCRC patients (*n* = 54) and healthy controls (HCs) (*n* = 48).

	mCRC	HCs	*p*-Value
Age (%)			
≥65	35 (61.4)	22 (38.6)	0.07
<65	19 (42.2)	26 (57.8)	
Sex (%)			
Male	39 (54.9)	16 (51.6)	0.66
Female	12 (20.3)	15 (48.4)	
Median Total EVs/µL (95% CI)	5264.0 (4123.0–6314.0)	2548.0 (2100.7–3051.4)	0.000003
Median CD133+ EVs/µL (95% CI)	52.6 (32.4–96.1)	18.4 (11.6–32.2)	0.0002
Median EPCAM+ EVs/µL (95% CI)	50.9 (38.6–67.2)	27.0 (19.3–42.0)	0.007

**Table 2 cancers-14-01357-t002:** Univariate Cox proportional hazards model predicting OS in the treatment-naive cohort (*n* = 33).

Variable	Univariate Analysis		Bootstrap Results (1000 Replicas)			
	HR (95% CI)	*p*	Bias	SE	95% CI	*p*
Total EVs ^a^	1.77 (1.24–2.54)	0.002	0.15	0.58	0.30 to 1.78	0.001 ^b^
EPCAM EVs ^a^	1.31 (0.94–1.84)	0.11	0.33	0.64	−0.03 to 2.02	0.32
CD133+ EVs ^a^	1.72 (1.24–2.39)	0.001	0.12	0.30	0.38 to 1.42	0.001 ^c^
ECOG PS						
1	1 [reference]					
0	0.15 (0.03–0.72)	0.02	−0.46	1.98	−8.37 to −0.08	0.003
Age (years)						
≥65	1 [reference]					
<65	0.48 (0.12–1.94)	0.30	−0.21	1.04	−4.02 to 0.68	0.13
No. of metastatic sites						
>1	1 [reference]					
1	0.95 (0.25–3.57)	0.94	−0.05	0.87	−1.73 to 1.41	0.93
Grading						
1–2	1 [reference]					
3	3.47 (0.37–32.4)	0.27	−0.18	3.91	−3.22 to 13.5	0.10 ^b^
Primary site						
Left-sided colon/rectum	1 [reference]					
Right-sided colon	2.75 (0.675-11.57)	0.17	0.01	1.81	−3.21 to 3.09	0.10 ^c^

^a^ continuous variable (log-transformed); ^b^ based on 927 samples; ^c^ based on 998 samples; abbreviations—HR: hazard ratio; SE: standard error; CI: confidence interval.

**Table 3 cancers-14-01357-t003:** Multivariate Cox proportional hazards model predicting OS in the treatment-naive cohort (*n* = 33).

Variable	Multivariate Analysis		Bootstrap Results (1000 Replicas)			
	HR (95% CI)	*p*	Bias	SE	95% CI	*p*
Total EVs	1.80 (1.06–3.09)	0.03	1.14	3.37	−0.10 to 11.30	0.01 ^b^
CD133+ EVs	1.67 (1.02–2.74)	0.04	0.73	3.21	−0.07 to 7.43	0.006 ^b^
ECOG PS						
1	1 [reference]					
0	0.06 (0.01–0.55)	0.01	−4.17	16.55	−36.16 to 1.65	0.02 ^a^

^a^ continuous variable (log-transformed); ^b^ based on 998 samples; abbreviations—HR: hazard ratio; SE: standard error; CI: confidence interval.

## Data Availability

The data that support the findings of this study are available from the corresponding author, D.B. (Davide Brocco), upon reasonable request.

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
