# Peer review of "Blood Circulating CD133+ Extracellular Vesicles Predict Clinical Outcomes in Patients with Metastatic Colorectal Cancer"

_cancers, 2022, doi:10.3390/cancers14051357_

Round 1

Reviewer 1 Report

In the introduction section, more references are needed to justify the use of CD133 and EpCAM as biomarkers of CRC EVs. 

In materials and methods section:

-the type of research made on human person should be precised. Is it a study made on a biologic collection ?

-in supplementary Table 1 , authors should explain what is "tumor grading".  

In the results section, in the Table 1, authors should invert the presentation of the results obtained for the 2 groups, in accordance with the previous presentation (Figure 1). 

In the paragraph 3.2. (dealing with the presentation of results obtained on overal survival in treatment naïve patients) is a little complex to follow. The same remark is applicable on the Table 3 and Table 4. Is it really necessary to present results obtained by univariate and multivariate analysis ? If the answer is yes, authors have to improve these results to be more clear. 

In the Figure 3, the name of the Y scale is missing for the box plot diagram (A) and histograms (B). 

Author Response

Manuscript ID: cancers-1608499

We would like to thank the Reviewers for careful and thorough review of our manuscript and for the thoughtful comments and constructive suggestions, which greatly help to improve the manuscript. Our point-by-point response to Reviewer’s comments (in italics) follows.

Point-by-point response to Reviewer’s comments

Reviewer 1:

1) In the introduction section, more references are needed to justify the use of CD133 and EpCAM as biomarkers of CRC EVs.

Reply: We thank the Reviewer for this comment. Following Reviewer’s suggestions, we added a comment and references at lines 96-98 of the revised manuscript to justify the use of CD133 and EPCAM for blood circulating EV characterization.

2) In materials and methods section:

-the type of research made on human person should be precised. Is it a study made on a biologic collection ?

Reply: Thank you for pointing this out. Peripheral blood samples were processed within 4 hours from venupuncture. EV analysis was not made on stored biological samples. We clarified this point at lines 121-122.

3) -in supplementary Table 1, authors should explain what is "tumor grading". 

Reply: We thank the Reviewer for this suggestion. A four-tiered grading system was used as recommended for histologic tumor grading evaluation in colorectal cancer by AJCC in the 8th edition of the TNM manual. We specified this in the foot note “a)” of the Supplementary Table 1 included in the revised supplementary materials.

4) In the results section, in the Table 1, authors should invert the presentation of the results obtained for the 2 groups, in accordance with the previous presentation (Figure 1).

Reply: We thank the Reviewer for this comment. As suggested by the Reviewer, we properly modified Table 1 by inverting the two lines referred to CD133+ and EPCAM+ EV concentrations. We modified the result section at lines 200-202 of the revised manuscript in accordance with the Figure 1 and the revised Table 1.

4) In the paragraph 3.2. (dealing with the presentation of results obtained on overal survival in treatment naïve patients) is a little complex to follow. The same remark is applicable on the Table 3 and Table 4. Is it really necessary to present results obtained by univariate and multivariate analysis ? If the answer is yes, authors have to improve these results to be more clear.

Reply: We thank the Reviewer for this comment showing that the paragraph 3.2 is complex to follow. We think that it is necessary to present results obtained by univariate and multivariate analysis. Specifically, univariate and multivariate analysis allow to correlate overall survival with EV concentrations treated as continuous numeric variables. The use of continuous variables may improve the statistical power of the survival analysis and help to better adjust the effect of confounding factors (Altman DG, et al. BMJ, 2006). We mentioned this point at lines 174-175 of the revised manuscript. Dichotomization of EV concentrations for Kaplan-Meier analysis followed the univariate cox proportional hazards regression analysis to improve the interpretation of results. According to Reviewer’s suggestion, we revised the text of the result section at lines 227-233  of the revised manuscript. We modified Table 2 and Table 3 of the revised manuscript to make them clearer.

5) In the Figure 3, the name of the Y scale is missing for the box plot diagram (A) and histograms (B).

Reply: We thank this comment. Labels for Y scales were added in the box plot  of Panel A and in the histogram of Panel B, which were included in the Figure 3.

Reviewer 2 Report

In their manuscript »Blood circulating CD133+ extracellular vesicles predict clinical outcomes in patients with metastatic colorectal cancer« the authors report on results of a study performed between 2017 and 2021 using a specific form of liquid biopsy concept in clinical setting.

In the introduction section they nicely explain potential clinical benefits of liquid biopsy concept in assessing and monitoring the metastatic disease as well as the molecular fundaments of extracellular vesicles (EV) use as a biomarker of liquid biopsy. In the methods section they provide clinical details of 54 included metastatic colorectal cancer (CRC) patients, 48 healthy control patients and technical details of detecting total and specific EV by flow cytometry. Statistical methods used in the study are also sufficiently described. In the results section they show that patients with metastatic CRC present higher blood levels of total and CD133+ circulating EV compared to healthy controls. Total and CD133+ circulating EV concentrations are associated with worse overall survival in treatment naive patients. In addition, CD133+ circulating EV are associated with reduced overall response rate to systemic treatment. The discussion is clear and straightforward. The results are nicely analyzed in the light of comparable published literature. The conclusions are sound and fair. The references sufficient, relevant and updated.

Author Response

Manuscript ID: cancers-1608499

We would like to thank the Reviewers for careful and thorough review of our manuscript and for the thoughtful comments and constructive suggestions, which greatly help to improve the manuscript. Our point-by-point response to Reviewer’s comments (in italics) follows.

Point-by-point response to Reviewer’s comments

Reviewer 2:

1) In their manuscript »Blood circulating CD133+ extracellular vesicles predict clinical outcomes in patients with metastatic colorectal cancer« the authors report on results of a study performed between 2017 and 2021 using a specific form of liquid biopsy concept in clinical setting.

In the introduction section they nicely explain potential clinical benefits of liquid biopsy concept in assessing and monitoring the metastatic disease as well as the molecular fundaments of extracellular vesicles (EV) use as a biomarker of liquid biopsy. In the methods section they provide clinical details of 54 included metastatic colorectal cancer (CRC) patients, 48 healthy control patients and technical details of detecting total and specific EV by flow cytometry. Statistical methods used in the study are also sufficiently described. In the results section they show that patients with metastatic CRC present higher blood levels of total and CD133+ circulating EV compared to healthy controls. Total and CD133+ circulating EV concentrations are associated with worse overall survival in treatment naive patients. In addition, CD133+ circulating EV are associated with reduced overall response rate to systemic treatment. The discussion is clear and straightforward. The results are nicely analyzed in the light of comparable published literature. The conclusions are sound and fair. The references sufficient, relevant and updated.

Reply: We sincerely appreciate the overall positive feedback of the Reviewer who believes that the “Statistical methods used in the study are also sufficiently described” and“The discussion is clear and straightforward”. We are happy that the Reviewer considers that “The results are nicely analyzed in the light of comparable published literature” and "The conclusions are sound and fair".

Reviewer 3 Report

This a new track to look for new potential biomarkers in mCRC. I would recommend to assess the outcome of circulating CD133+ extracellular vesicles regarding RAS and/or BRAF mutations, and the biologic used in combination with the chemotherapy.

Author Response

Manuscript ID: cancers-1608499

We would like to thank the Reviewers for careful and thorough review of our manuscript and for the thoughtful comments and constructive suggestions, which greatly help to improve the manuscript. Our point-by-point response to Reviewer’s comments (in italics) follows.

Point-by-point response to Reviewer’s comments

Reviewer 3:

1) This a new track to look for new potential biomarkers in mCRC.

Reply: We appreciate that the Reviwer finds that “This a new track to look for new potential biomarkers in mCRC.”

2) I would recommend to assess the outcome of circulating CD133+ extracellular vesicles regarding RAS and/or BRAF mutations, and the biologic used in combination with the chemotherapy.

Reply: We thank the Reviewer for this comment. According to Reviewer’s suggestion, we analyzed the correlation between OS and blood circulating CD133+ EV concentration according to K-RAS mutational status. We reported the Kaplan-Meier curves of K-RAS wild type and K-RAS mutated populations in Supplementary Figure 1 of the revised supplementary materials. We added a comment on this at lines 258-261 of the revised manuscript. We were not able to perform the analysis according to B-RAF mutation status because none of the patients included in the survival analysis presented a B-RAF mutation. The Reviewer suggested to evaluate the biologic agent used in combination with chemotherapy to stratify the analysis of outcomes. Unfortunately, we decided not to perform this analysis because of the small sample size of patient subgroups. Specifically, the group of patients with high level of CD133+ EV and treated with chemotherapy combined with anti-EGFR agents included only 3 patients whereas only 1 patient in the group of high CD133+ EV was treated with chemotherapy alone.

Reviewer 4 Report

Brocco et al. described an interesting study showing CD133+ extracellular vesicles being able to to predict clinical outcomes in mCRC patients. It is recommended that authors provide a thorough discussion regarding correlation of EVs with CTCs.

Author Response

Manuscript ID: cancers-1608499

We would like to thank the Reviewers for careful and thorough review of our manuscript and for the thoughtful comments and constructive suggestions, which greatly help to improve the manuscript. Our point-by-point response to Reviewer’s comments (in italics) follows.

Point-by-point response to Reviewer’s comments

Reviewer 4:

1) Brocco et al. described an interesting study showing CD133+ extracellular vesicles being able to to predict clinical outcomes in mCRC patients.

Reply: We appreciate that the Reviewer commented that “Brocco et al. described an interesting study showing CD133+ extracellular vesicles being able to to predict clinical outcomes in mCRC patients.”

2) It is recommended that authors provide a thorough discussion regarding correlation of EVs with CTCs.

Reply: We thank the Reviewer for this suggestion. A comparison between our findings on EPCAM+ EVs and CTCs has been provided at lines 370-379 of the revised manuscript.

3) Extensive editing of English language and style required

Reply: We thank the Reviewer for this comment. The manuscript has been extensively revised by two English experts.